# Academic Literacy and Student Diversity: Evaluating a Curriculum-Integrated Inclusive Practice Intervention in the United Kingdom

**Sara Calvo [1,\*], Luciano Celini [2], Andrés Morales [1], José Manuel Guaita Martínez [3] and Pedro Núñez-Cacho Utrilla [4]**

[1] Department of Business, International University of La Rioja, Avenida de la Paz, 137, 26004 Logroño, Spain; andres.morales@unir.net

[2] Business School, Middlesex University, Hendon, London NW4 4BT, UK; l.celina@mdx.ac.uk

[3] Faculty of Business, Valencian International University, Pintor Sorolla, 21, 46002 Valencia, Spain; josemanuel.guaita@campusviu.es

[4] Polytechnic School of Linares, University of Jaén, 23071 Jaén, Spain; pnunez@ujaen.es

**\*** Correspondence: sara.calvo@unir.net

**Abstract:** The sustainability of universities is based, among other aspects, on their ability to adapt to changes and the needs of students, an increasingly diverse population. In this sense, Academic literacy provision at universities tends to be centralized and to offer language support for general academic literacy purposes rather than language development that responds in a more nuanced way to the particular literacy needs of students' disciplines. Yet, in recent years, several studies have supported the integration of academic literacy into subject teaching outlining the principles of an inclusive model of academic literacy instruction. This paper draws on a theoretical framework developed by Wingate to evaluate a curriculum-integrated inclusive practice intervention in the United Kingdom with students from a first-year credit-bearing module at Middlesex University Business School. The study used a mixed methods approach that includes a literature review, secondary data, feedback questionnaire and a focus group to evaluate our teaching method and reflect on the collaboration of the team members to develop this inclusive pedagogical approach. The findings suggest that, on the whole, this intervention was perceived by both the module teaching team and students as positive, welcoming and often crucial for supporting undergraduate students into the disciplinary discourse of their subject of study. Yet, recommendations were made with respect to developing better guidelines for subject lecturers on how to deliver the integrated academic literacy as well as the importance of the participation of students, student learning assistants and graduate teaching assistants in the design of the intervention. This study contributes to the literature on inclusive practice intervention and pedagogical approaches to integrating academic literacy into subject teaching for a diverse student population, contributing to the social sustainability of the universities.

**Keywords:** academic literacy; curriculum-integrated design; inclusive practice intervention; student diversity

## 1. Introduction

Overcrowding, globalization, internationalization and the policies related higher education, have developed a complex and diverse student population in different parts of the world [1–3]. In this environment, the sustainability of universities, understood as their survival in the long term, involves attending to the diversity of their students and being inclusive. Students, entering universities, regardless of their linguistic, cultural and educational backgrounds, struggle to cope with the academic

communicative competence demands of their degree programs. In fact, most of the universities are implementing some form of in-sessional academic literacy support to develop their students' language proficiency [4].

The provision of academic literacy in universities tends towards centralization. Thus, linguistic support is being offered for general academic literacy purposes rather than language development, which responds in a more nuanced way to the particular literacy needs of student disciplines. However, in recent years, few studies have supported the integration of academic literacy into the teaching of subjects that describe the principles of an inclusive model of academic literacy instruction [5,6]. This is in line with Goal 4 of the Sustainable Development Goals (SDGs) that focused on inclusive education skills. In particular, objective 4.4 aims to "substantially increase the number of youth and adults who have relevant skills at different educational levels, including the university" [2]. Besides, this work is also aligned with objective 10 of the SDGs, which is reducing inequality within and between countries. Using inclusive policies in the university that allow students from many countries to access a quality university education reinforces equal development between countries.

Academic literacy refers to the ability to communicate competently in an academic discourse community (being competent in reading and writing on academic subjects). However, Porter [3] argued that this capacity requires the epistemological knowledge of the community, of the genres through which the community interacts, and of the conventions that regulate these interactions. This understanding of academic literacy has two main implications. First, academic literacy must be acquired by all students new in an academic context, whether native speakers or not. Second, outside of the community in which they operate, this literacy cannot be acquired. This means that experts from the discursive community must offer instruction and support to all students [4,5].

In the current university context in the United Kingdom, there is concern for long-term sustainability and there is a wide recognition of the need for student support in order to develop their academic literacy. Thus, an important body of literature has emerged that argues that academic literacy should integrate at the curricular level [4,5,7–10]. Specifically, in the 2017–2018 academic year, the total number of students enrolled in institutions of higher education (HE) in the United Kingdom amounted to 2.34 million. The growth of the sector has led to the configuration of increasingly diverse student populations on both issues, culturally and linguistically, reflecting the so-called "super diversity" [11].

In fact, there are currently 1.88 million students from the United Kingdom, 0.14 million from the European Union (EU) and 0.32 million from non-EU countries [12]. Therefore, there is increasing pressure on British universities to improve the retention, progress and performance levels of this population and to ensure that their academic offer recognizes diversity [12].

As a result, "student experience" has become a central concept in institutional planning. According to a study conducted in 2019 on the experience of freshmen in the United Kingdom, "lack of academic progress" is a key reason why a substantial number of undergraduate students drop out of their degree programs [13].

The integration of academic literacy in teaching subjects has been slow and, so far, the literature includes few examples of this, taken from the United Kingdom, Australia and South Africa [6,14–21]. These publications on educational initiatives of collaboration and specific discipline raise a series of questions. For example, there is no explicit explanation of the teaching methods and theoretical frameworks that support them [4,5]. Therefore, it is not clear to what extent literacy instruction is integrated within the curricula, or if on the contrary it is only a complement to them. In addition, little information is provided on the degree to which English language specialists for academic purposes (EAP), subject teachers and other related persons are involved in literacy. This lack of information makes it difficult for other institutions and individual professionals to learn from these examples and develop similar approaches [4,5].

## 2. Pedagogical Approaches to Academic Literacy: The Deficit and the Inclusive Collaborative Models

The most widely used approach in academic writing instruction is the Deficit or Skills model. This approach considers writing as a discrete skill and characterizes students as guilty of their "writing defects" [22,23]. Provision is usually carried out in generic workshops of extracurricular academic skills taught by centralized learning development units. These are usually located in the library or in student support services. The skills approach has predominated, as it offers a convenient and cost-effective reference route for "struggling" students. In addition, research on the effectiveness of generic skills workshops highlights that offering students opportunities and successful and enriching adaptive help is correlated with academic success [23–26]. However, generic workshops only capture superficial features of writing and do not address the "pluralistic nature of academic literacy" [27]. Therefore, they neglect the link between epistemology, discipline and language [13,27–29]. By separating writing from its disciplinary context, the skills approach places literacy on the periphery of university study. As a consequence, it contributes to the negative perception that both academics and students may have about their role in disciplinary teaching and learning [4,5].

The Inclusive Collaborative Model is the dominant critical framework for challenging the Skills or Deficit Model. This model emphasizes that literacies are socially located within their disciplinary contexts [28]. Within the disciplines there are very old and very differentiated literacy practices [29–32] in the construction of knowledge, such as research articles [33,34]. Similarly, the reproduction of knowledge has been performed through textbooks [33] and conferences [35]. Proponents of academic literacies affirm that the teaching of writing is most effective when it is within the discipline within which the literacy takes place [4,5,33,34,36,37]. In addition, the acquisition of academic literacy is an incremental process that requires frequent feedback on its development [38]. Therefore, academic literacy, an integral part of disciplinary thinking, is possibly more effective when integrated into the disciplinary culture and delivered longitudinally, using an inclusive and sustainable approach integrated into the curriculum.

Studies of literacy integration in the literature have typically focused on scheduled and specific academic writing interventions that are planned and delivered by teachers of the subject [5,12,39]. These are taught jointly by both professors of the subject and specialists in academic writing [5,40–42]. A common factor in these studies is the importance of collaboration between specialists in academic writing and subject teachers [13,40,43–45]. In fact, academic writing teachers, with their specialized knowledge in pedagogy and metalanguage, are important collaborative partners in the explicit articulation of the different disciplinary ways of building knowledge [13,43–45]. However, perfecting collaborative approaches to integrate academic literacy into subject teaching can be a gradual process. A study conducted in 2002 explained that, in the first instance, literacy specialists and subject teachers generally cooperate in the design of academic literacy materials that are integrated with the teaching of subjects [46]. Over time, this cooperation can lead to a close collaboration in the design of materials and, finally, to teaching as a team the specific academic literacy sessions of the subject.

The key benefit of developing a close collaboration is that writing is placed at the center of disciplinary learning and teaching, providing the best context to identify and address the specific difficulties students have when beginning disciplinary discourse [46].

In order for academic literacy to be integrated into higher education curricula, key stakeholders must be persuaded of the value and feasibility of systematic approaches integrated into the curriculum [45]. A starting point is that the institution's literacy officers facilitate a better appreciation by all academics of the complexities of social and writing practices in their community and offer them a sustainable model of literacy integration [27]. Wingate [4,5] defends an inclusive model of academic literacy instruction, advocating the adoption of the language of socialization and sociocultural theory as analytical frameworks for interpreting both academic literacy instruction and the systematic and gradual mastery of students from various disciplines, as well as the social and academic background of the defining conceptual basis of university disciplines. For Wingate [4], academic literacy is the ability

to communicate competently in an academic discourse community. This study, as discussed later in the methodology section, used a framework developed by Wingate [5] for the curricular interaction of academic literacy (see Table 1).

**Table 1.** Framework for Curriculum-Integrating Academic Literacy.

| Location | Delivery | Collaboration | Focus | Materials | Participation |
|---|---|---|---|---|---|
| Timetabled, credit-bearing (assessed component of content modules) | Subject lecturers; English for Academic Purpose (EAP) teachers | Input/advice from EAP teachers | Literacy conventions; Genres; Text features; Language for the creation of meaning and knowledge | Subject-Specific (Text tasks directly linked to classroom content) | Fully inclusive (Part of regular teaching, learning and assessment) |

Source: Wingate (2015:60).

## 3. Context and Methods

### 3.1. Study Context: Middlesex University Business School in London

The Middlesex University Business School has a worldwide presence on the university's campus network in London, Dubai, Mauritius and Malta. Currently, only on the London campus, there are students of more than 130 nationalities studying programs at the Business School. The university prides itself on the diversity of its student population. Its mission is to provide a global education that celebrates diversity while ensuring inclusion. In this way it will become a sustainable university. For almost a decade, the University Student Development Unit (LDU) has been collaborating with the staff of the subjects of the entire institution to incorporate academic literacy instruction in the teaching of the subject [47]. This support has normally been that members of the LDU have given ad hoc scheduled sessions on academic writing several weeks before deadlines. However, in 2013, in a review of the Business School programs, it was decided that academic literacy should be systematically integrated into the Business School curriculum. After several months of negotiations, it was agreed that the LDU team would co-design integrated literacy instruction with subject tutors from all departments of the school. The mentality at the faculty level has changed markedly and now the LDU has a central role in the curriculum of the Business School. In the case of the school's largest degree program, BA Business Management, academic literacy would be integrated into a central module in each year of study (first, second and third year).

Our study reports on the literacy intervention integrated in the curriculum in the first-year undergraduate module, with more than 300 students: HRM1004—"Management organizations", in the business management program of the University of Middlesex, London. Integrated literacy was taught in the form of classroom activities and online activities of own access. The materials were co-designed by the LDU tutor and the module leader. Students attended a mandatory 2-hour weekly workshop (around thirty students per workshop) as part of their first-year module, held between October and May 2016. In these sessions, the professors of the subject explored the theory and practice related to entrepreneurship, leadership and management, and organizational behavior. The tutors integrated 6 academic writing activities in class that were directly relevant to the content of the subject and the essay assignment, which was the main written evaluation of the module and represented 50% of the final grade (see Table 2).

The sixth and final literacy session was planned as a consolidation of previous activities to prepare for the presentation of the essay. The pedagogy employed and the integration planning were based on the literature on the integration of academic literacy in the teaching of subjects [12,37,39], but aligned within the Wingate framework [5] for integrated academic literacy in the curriculum.

**Table 2.** Description of in-class activities and self-access online activities.

| In-Class Activities (Delivered by Subject Tutors) | Description |
|---|---|
| (1) Assessments at university | An introduction to university assessment generally, with a focus on the assessments in the first year of study. This session outlined a clear rationale for the integration of academic literacy in the module. |
| (2) Reading critically | Students explore differences between a journal article (knowledge construction) and a textbook chapter (knowledge reproduction) from the module-reading list, both texts relate to the topic of 'personality'. |
| (3) Learning Techniques | Students reflect on their prior learning experiences and relate them to their learning on the module. |
| (4) Planning and Structuring your Essay | Using models of good practice from student assignments in previous years, students explore the genre features of essay writing and discuss, at a conceptual level, how they might structure their essays. |
| (5) How to paraphrase, cite and reference | Students explore the features of effective paraphrasing and summarizing and build an understanding of the conventions of the Harvard system of referencing |
| (6) Literacy session | A consolidation of previous activities to prepare for the presentation of the essay |
| **Self-Assessed Online Activities (compulsory)** | **Description** |
| (7) Time Management Questionnaire | Introductory online activity designed to introduce the students to the online materials interface and allows them to reflect on their previous learning experience. |
| (8) Reading at the University | Additional self-access online activity linked to in-class reading activity. |
| (9) Learning style questionnaire | Additional self-access online activity linked to in-class learning techniques activity. |
| (10) Referencing tutorial | Additional self-access online activity linked to in-class referencing activity. |

Source: compiled by the authors.

In applying the Wingate framework [4,5], the literacy component focused on literacy conventions, genres, text characteristics and language for meaning creation. Everything was directly related to the content and evaluation of the subject Business Management. Unlike other previous studies on the integration of academic literacy in the context of the United Kingdom, our study involved not only the collaboration between writing instructors and subject tutors, but also included the contribution of the postgraduate teaching assistant (GTA) and student learning assistants (SLAs) in the module. This is an additional aspect to the framework developed by Wingate. Student learning assistants are other experienced students who have previously been identified by their academic tutors as highly motivated and capable students. They work in conferences, seminars, workshops and small group sessions to help students in their learning. Graduate teacher assistants are graduates of Middlesex University who work to assist academic staff in providing additional support for students, such as face-to-face sessions, online support and assistance to academic staff in the development, production and delivery of materials of the program and the collection of data and information from programs and modules.

All integrated activities were directly linked to the criteria for qualifying the trial (see Table 3). An additional 30% of the final grade consisted of two online tests (in December and April), which assessed students' knowledge of the content of the classroom and independent study. The remaining 20% was participation, which included academic literacy activities online, in class and self-access.

**Table 3.** Linking the essay marking criteria with academic writing activities.

| Criteria | Excellent +70% | Good 60%–69% | Average 50%–59% | Pass 40%–49% | Fail −39% |
|---|---|---|---|---|---|
| Introduction: your introduction provides a clear idea of what your essay will be about, what theories will be presented and what structure the essay will take **(4)** | | | | | |
| Theories are relevant and considered appropriately to answer the question. Show good understanding of theories and reading related to the topic **(2, 3, 8, 9)** | | | | | |
| Evidence and Use of Research: Academic sources are used appropriately to support argumentation **(2, 5, 6,10)** | | | | | |
| Appropriate use of academic writing **(5, 6)** | | | | | |
| Focused on question set **(1, 4, 6)** | | | | | |
| Citations and references are used adequately **(5, 10)** | | | | | |
| Conclusion: there are logical arguments and ability to respond to the main question **(4, 7)** | | | | | |
| *General Comments* **Final Grade** | | | | | |

Source: compiled by the authors.

As seen in Table 2, the interactive online materials were created as a follow-up of classroom activities, providing students with more independent learning opportunities, facilitating both information acquisition and knowledge building [48]. It was also considered that students would welcome the integration of technology into the learning environment [42], as several studies have shown that students believe they benefit from the inclusion of technology in their learning [49] and "They appreciate the contributions that technology can make to improve their undergraduate education" [50]. The software used for online self-access materials was "Articulate Story Line", a program designed for non-expert technologists to develop interactive materials for teaching and learning.

*3.2. Methodology: Participants, Procedures and Data Analysis*

The study was conducted in four phases for twelve months. In the first phase, a literature search was conducted to obtain a common understanding of the conceptualizations and approaches of academic writing in higher education. This served to design the research methodology. In the second phase of the study, an analysis of the secondary data of student participation in online activities in class and self-access was carried out, along with their qualifications in their academic essays. A scale of 1%–100% was used, where 40% is a pass and 70% is a first or distinction. We presume that student participation in online writing activities, both in class and freely accessible, is associated with their grades in their academic essay.

Of the 324 students in the module, 166 participated in the study. The sample surveyed was composed of 88 women and 78 men, with an age of 18 to 45 years. 38% of the sample were mature students (over 21 years old). More than half (53%) of respondents were residents of the United Kingdom, 29% were residents of the EU and 12% did not belong to the EU. Only 6% of the sample did not answer the question about their residence. In relation to the "ethnic" dimension of the sample of respondents, 34% of the students were white, 31% Asian and 28% African black from the Caribbean,

followed by 2% Arab and 2% from other ethnicities. This indicates the diversity of the cohort and is representative of the wider student population of the University of Middlesex.

During the third phase of the study, we conducted a feedback questionnaire with 166 students (the same sample as in Phase 2). A 5-point Likert scale (1 = strongly agree and 5 = strongly disagree) was used to assess the effectiveness of online and classroom access. The questionnaire was completed in class and the students voluntarily completed it. First, students completed a data form designed to obtain information about their gender, ethnicity, residence and age. The remaining questions were related to whether literacy content in class and online activities was presented at an appropriate level, if there were clear instructions, if activities were increasing interest in writing and if these activities were helping students with their academic studies and writing development An open space was provided in the questionnaire where students were asked to write about their general perceptions and attitudes about integrated literacy content in class and online, as well as recommendations for future cohorts.

In the fourth phase of the study, a group discussion was held with the module's teaching team, which included the module leader, two subject teachers, an LDU academic writing teacher, a GTA and an SLA. The focus group schedule included open-ended questions and participants answered questions related to students' academic literacy and their participation in online class and self-access activities. The six participants were asked to take turns discussing the questions. The focus group was used to complement and contrast the information obtained from Phases 1, 2 and 3.

Ethical approval was obtained from the University of Middlesex and the participants, which included informed consent, confidentiality and "responsible" research practice. The information collected in the focus group was recorded and transcribed. The statistical package for the social sciences (SPSS) was used to analyze the relationship between student participation in online activities and in self-access class, and grades in the academic essay and student perceptions taken from students' feedback questionnaire data. Thematic analysis was used for qualitative data to identify topics through an iterative process of comparison and juxtaposition in a smaller number of higher order categories. The key issues were identified from the feedback questionnaire and the focus group and were refined as the analysis evolved. This analysis was a recursive rather than linear process that involved a constant round-trip movement between the entire data set, the issues and extracts of the data we identified and the data produced. The qualitative results were organized into four main themes: (1) the experiences of the students and the teaching team, (2) the preference for certain activities, (3) access to additional writing resources and (4) the experience of teaching academic writing in collaboration.

## 4. Research Findings and Discussion

### 4.1. Experiences of the Students and the Teaching Team

In general, students' comments indicated that they saw both classroom and online activities in a very positive way:

*'The activities helped me to build my self-confidence with writing', 'helpful activities', 'I strongly recommend them', 'they were enjoyable and interesting'.*

Of the 166 students who answered the feedback questionnaire, 75% felt that both their own online access and class activities were enjoyable, with only 16% of the sample disagreeing and 9% of the sample neither agreeing nor disagreeing. In addition, of the 166 students who participated in the feedback questionnaire, 77% stated that class and online activities were performed at the correct level. Only 18% of the sample did not agree, and a small number of students (5%) did not agree or strongly disagreed. Generally, students felt that the integrated activities had contributed significantly to their academic literacy development [4,10–13]. As one student pointed out:

*"The online and in-class activities were great, so clear and simple to understand each aspect. We learn how we could successfully write the essay; this really helped me in this module as well as in my other modules. It was just amazing."*

However, a small minority did not find the writing activities as useful or necessary for their academic literacy development and requested that these activities be additional learning material available outside the module. For example, a mature student commented:

*"On occasions, I felt that some of the activities were the wrong level. While I appreciate that the course must cater for all students, I expected more content in the module, rather than instructions on how to write essays."*

In this sense, this small number of students perceived academic literacy as a separable skill, distinct from disciplinary teaching and learning [22]. However, most of the students felt the benefits of having literacy activities integrated in the teaching of the subject [4,5,33,34,36]. Respondents in the discussion group were also interested in emphasizing the beneficial aspect of students participating in online and classroom writing activities. As one of the professors of the subject observed:

*"Well, I can see that students have benefited from these activities, their academic essays have a better quality than of students from previous years."*

Another example of this is found in a statement made by the GTA who commented:

*"I think we all agree that there is a need to develop writing skills for 1st year students and the idea to incorporate writing within the workshops is a really good idea because it introduced them with writing at university and gives them confidence in developing skills."*

This supports the well-established argument that today's students need support with their academic literacy development and that the provision must be integrated into the subject teaching [4,5,10–12]. However, the SLA that participated in the focus group discussion highlighted the differences between the expectations of mature students and their younger counterparts:

*"We need to take into account that there is a clear difference between mature and non-mature students as the mature students find the academic writing activities unnecessary, but the others as well as international students find this crucial, it is very difficult to keep a balance as you have different type of students with different needs."*

Exploring the topic further, a subject lecturer replied:

*"Well, I think some students have benefited more and others less, but overall I can say that even mature students needed academic writing skills, after looking at the first submission of their essays I could see that they didn't understand the nature of the questions, even if they were mature, they didn't have the experience in academic writing."*

This supports the view that both systematic and inclusive models of academic literacy integration can benefit all the students [4,10–13].

One aspect not captured by the feedback questionnaire, but that was identified by the focus group discussion, was that sometimes, students underestimated the importance of the development of academic literacy and the complexity involved in the processes and practices involved in the academic essay production. Thus, one of the professors commented:

*"I think they don't understand the importance of what we teach, it looks a bit basic, they find this a waste of time, and I think they take it for granted that it is one of the most fundamental things to learn at university . . . they should appreciate that we are explaining this because it will help them at university and beyond."*

In this respect, the findings suggest that although subject lecturers understood the importance of academic writing in the curriculum, some students underestimated its significance. Following this and to explore the extent to which students benefited from the academic writing activities, a correlation Pearson's analysis was conducted looking at the relationship between students' participation in the in-class and self-access online activities and essay performance. As observed in Table 4, the Sig. (2-Tailed) values are 0.003 and 0.019. With this value at less than 0.01, we can conclude that there is a statistically significant correlation between students' participation in the self-access online and in-class activities and essay performance. These findings are consistent with previous studies that suggest that students who have participated in writing activities had made progress in their assessments [12,41]. However, the findings show a weak relationship between in-class activities and essay performance ($r = 0.228$, $p > 0.01$). Moreover, there is a weak correlation between the variables online activities and essay performance ($r = 0.183$, $p > 0.01$).

**Table 4.** Correlations.

|  |  | Essay Performance |
| --- | --- | --- |
| Online activities participation | Pearson Correlation | 0.228 ** |
|  | Sig. (2-tailed) | 0.003 |
|  | N | 166 |
| In-class activities participation | Pearson Correlation | 0.183 * |
|  | Sig. (2-tailed) | 0.019 |
|  | N | 166 |

** Correlation is significant at the 0.01 level (2-tailed); * Correlation is significant at the 0.05 level (2-tailed). Source: compiled by the authors.

### 4.2. Preference for Certain Activities

It is interesting to note that students rated the activities in class as more useful than the online activities of their own access. While 79% of the students agreed or strongly agreed that class activities helped them develop their academic literacy, only 66% of the sample did so with online self-access activities:

*"With the in-class activities we were able to learn more actively and also it kept us engaged with the tutor."*

*"The in-class activities were more useful than the online games. In fact, the online activities needed better instructions and there were some technical problems. Well, the layout was not very good as you had to scroll to the sides to see the questions."*

This is in line with Wingate and Dreiss [42], who suggested that, although online tools can offer useful development opportunities for students, face-to-face support is also needed. The results of the feedback questionnaire indicated that the vast majority of students spoke of clear instructions in academic writing activities, with a better response for classroom activities (76%) than for online activities (73%). This was also raised in the focus group discussion, where one of the professors argued:

*"I think the main problem is that with the online activities, students have a weekly commitment to work outside the classroom to complete the activities; that is why I think they don't like the online activities."*

From the comments of the interviewees, it became clear that they considered that some activities were more useful than others:

*"I have had a look at students' feedback and I can say that some students highlighted that there were several activities that were very useful, for example the workshop that was delivered by the tutor from the Learner Development Unit as well as the one about structuring your essay. Well, regarding the*

*online activities the activities that students liked most were referencing and citations and adequate vocabulary for academic writing."*

The results of the feedback questionnaire revealed that the class activities students considered most useful were: evaluation in the university (86%), planning and structuring of their essay (82%), learning (77%), teaching on how to cite and reference (71%), followed by how to use academic words and reading in college (67%) and how to paraphrase (60%). On the other hand, in terms of online activities, the activities that were considered most useful were: reference tutoring (70%), university reading (66%), learning styles (65%) and time management questionnaire (59%).

### 4.3. Access to Additional Writing Resources

Another emerging issue was the notion that students accessed different additional writing resources outside their study program. In particular, the LDU professor who participated in the focus group discussion mentioned that he noticed an increase in the students of the module who reserved academic writing tutorials in his department:

*"I notice from this module a considerable number of students were booking tutorials and it appears that in comparison with other 1st year business and management modules, we have more bookings, specifically after they got their feedback from their first submission. I don't mean hundreds of students but just I notice a considerable number of them which is good because it indicates to some extent that the LDU working on their modules and programs has an impact on engagement."*

This evidence shows that a significant number of students considered their academic literacy development as an integral part of the development of their disciplinary knowledge. They also recognized the LDU as a useful development resource for seeking self-directed help [24,25]. However, focus group participants highlighted some recommendations to further improve student participation. As the module leader noted:

*"I think we need to promote what we are doing. I don't see many people accessing the online activities, so maybe from module leader and tutors, to promote that side of learning and structure of the activities, and what they will gain from the activities. And if they see a rationale behind it they will be more motivated."*

Regarding online activities, the study suggests that the low participation of students was due to the low level of participation of teachers in online activities [42]. In addition, participants in the focus group discussion considered it crucial to reevaluate class and online activities in a participatory approach for the next academic year. They also emphasized the importance of including students in the design of academic literacy integrated in the curriculum in the module. One of the tutors of the module suggested that:

*"We need to include students in the design of the module, well we need to reflect on how to improve the activities, we need to include more activities on how to structure your essay, and help on how to get a good grade and language style, maybe less on assessments at university."*

### 4.4. The Experience of Teaching Collaboratively Academic Writing

Another interesting topic identified by the participants in the focus groups was the experience of subject teachers who teach academic literacy. During the academic year, an LDU writing teacher prepared the academic writing materials in collaboration with the module leader. The module teaching team then delivered the integrated materials following the instructions and guidance provided by the LDU. The discussion that arose was related to whether the professors of the subject were skilled enough to teach academic literacy:

*"I think X's [LDU staff member] session was good, this leads me to the idea that the integration of writing skills is very important but wonder whether we are the right people (module tutors) to teach*

*that topic, because yes, we have written essays, and we know what is required for academic writing but teaching is different and I wonder if it is better to get X or somebody else like him to run those sessions."*

Another subject lecturer responded to the issue slightly differently by commenting:

*"To be honest, I am quite happy to carry on like this, to keep the in-class activities because they are first-year students and they may agree that obviously people from the LDU can do it better than us, but these are first-year students and the material is going to be quite basic anyway."*

These findings demonstrate that refining collaborative approaches to integrate academic literacy can be a slow and heuristic process [46]. It also highlights the importance of close collaboration between writing instructors and subject tutors in the design and delivery of integrated literacy [13,40,44,45].

## 5. Conclusions

Sustainability in the study programs in universities involves addressing tasks such as inclusive academic literacy. This study provides evidence of the evaluation of an inclusive academic literacy intervention integrated into the curriculum [4] that combines multiple delivery methods (classroom and online activities).

In general, the integrated activities were perceived by both the staff of the subject and the students as positive, welcoming and often crucial to support the induction of university students in the disciplinary discourse of their subject matter [10,28]. Statistical data show that there is a relationship between student participation in activities and essay performance, although this relationship is weak. However, the study was limited to the first year of the program and, therefore, was only able to measure the early stages of initiation in disciplinary discourse, an incremental process that involves frequent comments on development [30].

Another interesting finding was the increased participation of students with academic literacy development opportunities outside their study program, suggesting that the integration of literacy at the curriculum level made students more aware of the importance of academic literacy in their learning in general and created greater opportunities for success seeking adaptive help [25,26,34].

The findings also draw attention to the need to reevaluate online activities in the classroom and self-access. Additionally, the possibility of redesigning the module in a participatory approach, using data from the questionnaire, student feedback and the comments and recommendations of the broader teaching team of the focus group discussion, must be considered. In fact, some participants recommended several improvements that could be made to writing activities. In particular, these recommendations were related to the importance of developing better guidelines for subject teachers on how to offer the integrated component of academic literacy, as well as advocating for student participation in module design [12,46].

In addition, the results demonstrate that perfecting collaborative approaches to integrate academic literacy by being a slow and heuristic process requires close collaboration between writing instructors and subject tutors in the design and delivery of integrated literacy [13,40,44,45].

Therefore, this study contributes to the literature on inclusive procedural and pedagogical approaches to integrate academic literacy into the teaching of the subject. Here, we demonstrate that inclusive and innovative ways of leaving no one behind can have significant repercussions for the teaching–learning process, for the student's experience and for the reputation of universities [1–3]. Based on the findings, it can be argued that there is a need to develop inclusive and sustainable collaborative teaching patterns and practices that can help subject teachers integrate academic literacy into their curricula and reflect on the importance of including a broader teaching team (for example, SLA and GTA) and students in the design of literacy integrated into the curriculum. These contributions are in line with the demands of the United Nations, which pay attention to the sustainable development in teaching at universities [51,52].

In addition to practical concerns, we believe that longitudinal research should be conducted to better assess the development of student academic literacy throughout the program's life cycle (for example, during the duration of their bachelor's degree program). More qualitative and quantitative studies should also be performed to review and compare the different levels of integration of academic literacy in different disciplinary areas of higher education in the United Kingdom and worldwide in order to identify the characteristics of 'good practice' so as to ensure an inclusive collaboration practice of the intervention.

**Author Contributions:** Conceptualization, S.C. and L.C.; Formal analysis, A.M.; Investigation, S.C. and L.C.; Methodology, S.C. and L.C.; Supervision, A.M., J.M.G.M. and P.N.-C.U.; Validation, J.M.G.M. and P.N.-C.U.; Writing—original draft, S.C. and L.C.; and Writing—review and editing, A.M., J.M.G.M. and P.N.-C.U. All authors have read and agreed to the published version of the manuscript.

**Funding:** This research received no external funding.

**Conflicts of Interest:** The authors declare no conflict of interest.

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
