# Peer review of "Academic Literacy and Student Diversity: Evaluating a Curriculum-Integrated Inclusive Practice Intervention in the United Kingdom"

_sustainability, doi:10.3390/su12031155_

Round 1

Reviewer 1 Report

Detailed recommendations in the (upended) text of review.

Author Response

Dear Reviewer, I would like to thank you for your valuable comments.

We have revised the paper to address fully these outstanding concerns. Full details of our response to each of the comments made are set out below and all changes made are highlighted in the text (yellow colour). As you can see, I have made significant changes to the paper to ensure that it is clearer the purpose of this study and the contribution.

If you require any further information, please do not hesitate to ask.

Yours sincerely

Dr Sara Calvo Martinez

Reviewer comments

Response

1. Author’s opinion vs literature knowledge

Several sections have been entirely rewritten to ensure clarity between author’s opinion and literature knowledge as well as research gap and contribution.

2. British Universities or broader

We have made changes to ensure it is clear in the paper where we refer to UK universities or broader

 3. Abstract and keywords

Done

Tables  

Done

Key definitions

We have included key definitions in particular the one related to academic literacy.

We (expressions)

We have deleted ‘we’ from different parts of the text. You can only find 6 times where we include the word ‘we’.

 Methodological Section

We have made changes and explain in more detail the data analysis section.

Findings and Discussion Section

We have made changes accordingly. In particular. we have deleted Part 5 (Teaching Team Perceptions) and include the information in the theme above. We have included the word collaboratively in the findings.

Conclusion Section

We have changed conclusions taking into account the reviewer comments.

Reviewer 2 Report

Dear authors,

Firstly, I would like to thank you for submitting the paper entitled “Sustainability of Inclusive Education: Reflecting and Evaluating a curriculum-integrated academic literacy intervention in the United Kingdom” submitted to Sustainability.

In my opinion this paper is out of the scope of the journal. An alternative, decentralised model of language support implemented at Middlesex University Business School is analyzed. The relationship between the paper and sustainability is very scarce. In my opinion this paper should be published in a journal directly directed with education.

In my opinion the article is well structured. In paragraphs 1 and 2 the you introduce the research that has been done. The objectives of the research are defined too.

In section 3 the methodology used is described. The authors refer to Appendix 1. Nevertheless the Appendix 1 is not in the manuscript.

In my opinion, sections 4 and 5 should be unified into a single section.

Authors should introduce a section with the authors' contribution, declaration of conflicts of interest, acknowledgements and possible sources of funding.

Minor revisions:

The format of the section and sub-section titles does not correspond to that proposed in the instructions to authors. The format of the references in the text and the list of references does not coincide with that of the journal. The size of tables 1 and 3 does not correspond to the proposed size of the journal. The format of tables 2 and 4 does not conform to the format of the journal. It would be interesting to introduce the text of the footnotes in the main text.

In my opinion this paper could be published with minor revisions in other MDPI journal: Education Sciences.

Author Response

Dear Reviewer,

I would like to thank you for your valuable comments.

We have revised the paper to address fully these outstanding concerns. Full details of our response to each of the comments made are set out below and all changes made are highlighted in the text (yellow colour).

If you require any further information, please do not hesitate to ask.

Yours sincerely

Dr Sara Calvo Martinez

Reviewer comments

Response

Out of the scope of the Journal

We have justified in the paper why we believe this paper is relevant for this Journal. We consider this paper relates to the Sustainable Development Goals (SDGs) that focused on inclusive education skills. See highlighted in yellow.

Methodology (Appendix 1)

We have deleted Appendix 1 (added Table 1 instead)

 Sections 4 and 5 should be unified

Done

Author’s contribution, Declaration of conflicts of interest, acknowledgements

Done

Sections title, tables, references and Footnotes

We have changed the title, tables, references and include footnotes in the main text.

Reviewer 3 Report

The paper is relevant, the topic of the evaluation of the academic literarcy intervention is actual. The focus is on feedback from students and teaching team concerning the academic support.

I make some suggestions:

2-4 Title

-not correctly describes the paper examination, REFORMULATE!!! 

the environment is inclusive and your general objective can be the pedagogical sustainability , but here, on this paper, your focus is on description and evaluation of the program you presented a case study, a model of language support

13-27 Abstract

is not a logical connection between the problem formulation and conclusions is missing the study methodology presentations  18-20 REFORMULATE!!! (your demonstration through this paper is other) keywords - rethink 

138-191 this is not a study methodology!!! MOVE on 2.2.

this is a description of the programme 

240 The evaluation is also about negatively of the programme, not just about the positive experience. I suggest you `Students and the teaching team experiences ...`

241-248 how about the 16% of the sample disagreeing???

291 change 5. in 4.2.

300-314 table 4. is not correct presented!!! 

you must to give a title Pearson correlation values must be between -1 to 0 or 0 to 1 (not 28 or 183).  is important the positively or negativity of values  * and **???

315 change in 4.3.

347 change in 4.4.

376 change in 4.5.

395 change in 5. 

405-410 reformulate concerning your findings on 347-375!!!

432- Refereces!!! CHANGE!!!

Is a discrepancy between `on text` and `end text` references. 

Where is 1 or 2 or.... on the body-text???

Author Response

Dear Reviewer,

I would like to thank you for your valuable comments.

We have revised the paper to address fully these outstanding concerns. Full details of our response to each of the comments made are set out below and all changes made are highlighted in the text (yellow colour).

If you require any further information, please do not hesitate to ask.

Yours sincerely

Dr Sara Calvo Martinez

Reviewer comments

Response

Title

We have changed the title to ‘Academic Literacy and Student Diversity: Evaluating a curriculum-integrated inclusive practice intervention in the United Kingdom’

Abstract and keywords

We have changed the abstract and keywords

 138-191

Study context (rather than methodology)

240

We have changed as suggested by the reviewer from ‘Positive Experiences’ to ‘Students and the teaching team experiences…’

241-248

How about the 16% of the sample disagreeing? There is no data to discuss further. 

291

Change 5 im 4.2 (Done)

 300-314

Table  4. Change table values (-1 to 1) Done

315,347,376,395

Done

405-410  

Refomulate your findings 347-375. Done

References

Done

Round 2

Reviewer 1 Report

The Authors significantly improved the quality of the study presentation. In particular, the new version makes it easier to distinguish between knowledge known from the literature and the results of the authors' research.

The current version may be published.

Author Response

Dear Reviewer,

Please, find attached the response to your comments. Thank you very much for your valuable feedback. 

Best wishes

Sara

Reviewer 2 Report

Dear authors,

Thank you very much for your effort to try to frame the article in Sustainability magazine. One of the keys to the Sustainable Development Objectives is that it is a comprehensive agenda. With what other objectives is the work presented related?

The lack of relation with the subject of sustainability is shown in the references used in the discussion of the article.

In my opinion, for the article to be published in the journal Sustainability, the relationship must be throughout the article and the authors should focus the discussion on how the proposal contributes to the sustainability of education.

Minor revisions:

Article format is not according journal instructions.

The format of the section and sub-section titles does not correspond to that proposed in the instructions to authors.

The format of the references in the text and the list of references does not coincide with that of the journal.

Figure 1 captions has a reference that is not in journal format.

In my opinion this paper could be published with minor revisions in other MDPI journal: Education Sciences.

Author Response

(The authors gave the same response as above.)

Reviewer 3 Report

Congratulation!

Author Response

(The authors gave the same response as above.)
